# SmilesFormer: Language Model for Molecular Design

## Abstract

The objective of drug discovery is to find novel compounds with desirable chemical properties. Generative models have been utilized to sample molecules at the intersection of multiple property constraints. In this paper we pose molecular design as a language modeling problem where the model implicitly learns the vocabulary and composition of valid molecules, hence it is able to generate new molecules of interest. We present SmilesFormer, a Transformer-based model, which is able to encode molecules, molecule fragments, and fragment compositions as latent variables, which are in turn decoded to stochastically generate novel molecules. This is achieved by fragmenting the molecules into smaller combinatorial groups, then learning the mapping between the input fragments and valid SMILES sequences. The model is able to optimize molecular properties through a stochastic latent space traversal technique. This technique systematically searches the encoded latent space to find latent vectors that are able to produce molecules to meet the multi-property objective. The model was validated through various *de novo* molecular design tasks, achieving state-of-the-art performances when compared to previous methods. Furthermore, we used the proposed method to demonstrate a drug rediscovery pipeline for Donepezil, a known acetylcholinesterase inhibitor.

## 1 Introduction

Generative models play a major role in discovering and designing new molecules, which is key to innovation in in-silico drug discovery (Schwalbe-Koda & Gómez-Bombarelli, 2020). There is a vast amount of molecular data available, therefore generative models should be able to learn concepts of valid and desired molecules using a data-centric approach. However, due to the high dimensionality of the molecular space (Probst & Reymond, 2020) and the substantial number of data samples, it is still challenging to traverse the space of valid molecules that satisfy desired objectives of *de novo* molecular design. Other challenges that have been identified for *de novo* molecule generation task include the reliance on a brute-force trial and error approach to searching for hit compounds, the difficulty in designing an effective reward function for molecular optimization, sample efficiency due to online oracle calls during molecule optimization (Fu et al., 2022). Addressing these challenges is still an active area of research with various approaches such as Reinforcement Learning (Zhou et al., 2019)(Jin et al., 2020b), Genetic Algorithms (Wüthrich et al., 2021), Variational Auto-Encoders (VAEs) (Kusner et al., 2017) and Generative Adversarial Networks (GANs) (Schwalbe-Koda & Gómez-Bombarelli, 2020; Prykhodko et al., 2019).

In this work, we employ a Transformer-based language model (Vaswani et al., 2017) to encode a molecular latent space by generating valid molecule sequences from fragments and fragment compositions. The encoded latent space can then be explored to generate molecules (represented by SMILES strings (Weininger, 1988)) that satisfy desired properties.

Similar to Gao et al. (2022) who addressed the problem of synthesizability by modeling synthetic pathways within the molecule design pipeline, we leverage data to minimize costly experiments in downstream generation and optimization tasks using a fragment-based approach that associates synthesizable building blocks with target molecules during training. Our model, however, introduces an online fragmentation approach, removing the need to create a separate fragments dataset while learning the intrinsic relationship that makes up the formation of a valid SMILES string. This is essentially an approach for learning the SMILES language model.

While SMILES has been seen to be less informative compared to other forms of molecule representation, we argue that it is simple and easy to follow as it is a linear walk through the molecular structure. Also, we see the non-canonical property as a form of data augmentation which has been shown to benefit the training of generative models (Coley, 2021). We explored this idea by using only non-canonical SMILES as input to our model and teaching the model to generate canonical SMILES.

Our contributions are summarized as follows: (1) We propose an approach for learning efficient representations of the molecular space using molecule fragments. Our fragment-based training pipeline constrains the model to learn the building blocks necessary to generate stable molecules while also meeting multi-property objectives. (2) We present an optimization strategy that efficiently traverses the molecular space with flexible parameterization. (3) We demonstrate a *de novo* molecular optimization practical use case with a rediscovery pipeline of an established acetylcholinesterase inhibitor.

## 2   RELATED WORK

The majority of generative models focus on the generation of valid molecules with desired properties (Gao et al., 2022), however with great reliance on domain knowledge in medicinal chemistry to help guide the generation process. For example in Jin et al. (2018), an approach based on VAEs (Kingma & Welling, 2014) generates a molecular graph by first generating a tree-structured scaffold over chemical substructures. This involved building a cluster vocabulary that increases as the size of the considered chemical space increases. Other more recent state-of-the-art approaches are based on genetic algorithms through expert-guided learning (Ahn et al., 2020; Nigam et al., 2021; Wüthrich et al., 2021), differentiable scaffolding tree aided by a graph convolutional knowledge network (Fu et al., 2022), and VAE reinforcement learning with expert designed reward function to explore encoded latent space (Zhavoronkov et al., 2019).

Our work is motivated by Fabian et al. (2020) who made an earlier attempt at using a Transformer-based language model applied over a vast amount of data to learn the representation of molecules for domain-relevant auxiliary tasks, however, did not explore *de novo* molecular design. Defining molecular design as a language problem enabled us to move away from heavy reliance on expert knowledge and allows us to choose a method that can scale over a massive amount of available datasets. Intuitively, we see fragments as similar to words or phrases in a language sentence. Hence, learning the relationship between fragments and full sequences means a full sequence can be generated from fragments or even a combination of fragments. The most similar works to this idea use molecule fragments as the building block for molecule generation pipelines (Polishchuk, 2020), showing that fragment-based approaches are in-between atom-based approaches and reaction-based approaches. While the approach utilizes a database of known compounds to perform chemically reasonably mutation of input structures, ours relied on an online stochastic fragmentation process based on a retrosynthetic combinatorial analysis procedure (RECAP) (Lewell et al., 1998) provided by the RDKit library (Landrum, 2022). This fragmentation process is only used during training and fragments are not stored for later use. Another method (Firth et al., 2015) coupled a rule-based fragmentation scheme with a fragment replacement algorithm to broaden the scope of re-connection options considered in the generation of potential solution structures. Fragments were used optionally in Nigam et al. (2021) to bias the operators in the genetic algorithm pipeline. The effect of this optional bias, however, was not reported.

Similar to the REINVENT (Blaschke et al., 2020), which was developed as a production-ready tool for *de novo* design, applicable to drug discovery projects that are striving to resolve either exploration or exploitation problems while navigating the chemical space, we adopt a transfer learning approach that utilizes a pretrained generative model as a prior to further train a smaller set of compounds which are relevant for the desired outcome in downstream tasks.

To evaluate generative models for molecular design, distribution benchmarks and goal-directed benchmarks (Brown et al., 2019; Polykovskiy et al., 2020) have been proposed. However, since the task of molecular design is usually related to specific targets, earlier proposed benchmarks do not fully represent how a generative model is utilized in the drug discovery pipeline. We observe that recent works have been more focused on multi-objective property optimization as this is more practically useful. Aided by standardized libraries like TD Commons (Huang et al., 2021), recent works from Jin et al. (2020b); Nigam et al. (2021); Fu et al. (2022), and Gao et al. (2022) evaluated their models

using optimization for activities such as JNK3 and GSK3$\beta$ (Li et al., 2018) in combination with QED (Bickerton et al., 2012) and Synthetic Accessibility (Ertl & Schuffenhauer, 2009). In this work, we further evaluate our approach by simulating a known drug discovery pipeline to demonstrate how our model can be utilized in a more practical setting.

## 3 METHOD

### 3.1 SMILESFORMER MODEL

Our generative pipeline is a Transformer-based VAE, trained to obtain non-deterministic outputs of a molecular sequence. In our implementation, we constrain the latent representation between the encoder and decoder on the standard Normal distribution to achieve the variational auto-encoding. More details on the Transformer model is in section A.1, and our model's (SmilesFormer) architecture and molecule optimization overview is shown in figure 1.

The model accepts input sequence tokens $s_1, ...., s_K$, which can be from full sequences, partial sequences, fragments, fragment compositions or an unknown fragment (Figure 2), combined with query sequence tokens $q_1, ...., q_T$, are mapped to probability distributions of output tokens $P(x_1), ...., P(x_T)$. The encoder encodes the input sequence to obtain a latent space representation $\boldsymbol{z}$ constrained on the standard Normal distribution $p(\boldsymbol{z}) = N(0_{d_z}, I_{d_z})$, such that $\boldsymbol{z} \sim f(S, p(\boldsymbol{z}))$, where $d_z$ is the dimension of the latent space. The decoder $g(\boldsymbol{z}, Q)$ then output tokens probability:

$$P(X|Q, \boldsymbol{z}) = \prod_{t=1}^{T}(x_t|X_{<t}, \boldsymbol{z}) \tag{1}$$

where $Q \sim X_{<t}$. In a greedy approach, the model obtains the output token $T_t = argmax(P(X))$. The parameters of the encoder $\theta_E$ are trained to minimize the Kullback-Leibler (KL) $D_{KL}(\cdot||\cdot)$ divergence between the latent space distribution and a Normal distribution:

$$L(\theta_E) = D_{KL}(p(\boldsymbol{z}|Q)||p(\boldsymbol{z})) \tag{2}$$

The parameters of the decoder $\theta_D$ are trained to minimize the Negative Log-Likelihood:

$$L(\theta_D) = -\sum_{i=1}^{N} \ln P(x_i = X_i|Q, \boldsymbol{z}) \tag{3}$$

For weight variables $\omega_E$ and $\omega_D$ used to balance the encoding objective and the sequence reconstruction objective respectively, the final objective function is then given by,

$$L(\theta_E, \theta_D) = \omega_E \cdot L(\theta_E) + \omega_D \cdot L(\theta_D) \tag{4}$$

### 3.2 MOLECULE OPTIMIZATION

The objective of molecule optimization is to take a molecule $X$ as input and output a new molecule $X^*$ that is similar to $X$ and has more desirable properties than $X$. We formulate a latent space manifold sampling strategy based on Mueller et al. (2017) which operates under the standard supervised learning setting, assuming the availability of a dataset $D_n = (x_i, y_i)_{i=i}^{n} \sim Pxy$ of sequence-outcome pairs. The distribution $Px$ is from the outputs of our learned decoder and $Py$ is the distribution of molecule scores. By fitting models to $D_n$, we seek to obtain a revised sequence:

$$x^* = argmax_{x \in C_X} E[Y|X = x] \tag{5}$$

where $X_C$ is a set of valid sequences or revisions, that is molecules that satisfy the multi-property objective. The relationship between $X$ and $Y$ is parameterized by a simple feedforward network. Given an input sequence $X$ to be optimized, we first compute its latent space representation $z_0 = E(S, p(z))$ using our trained encoder. Using the gradients of the local optimum of the simple feedforward network, we obtain an updated latent variable $z^* \sim z + \alpha \times grad(z)$, where $\alpha$ is a

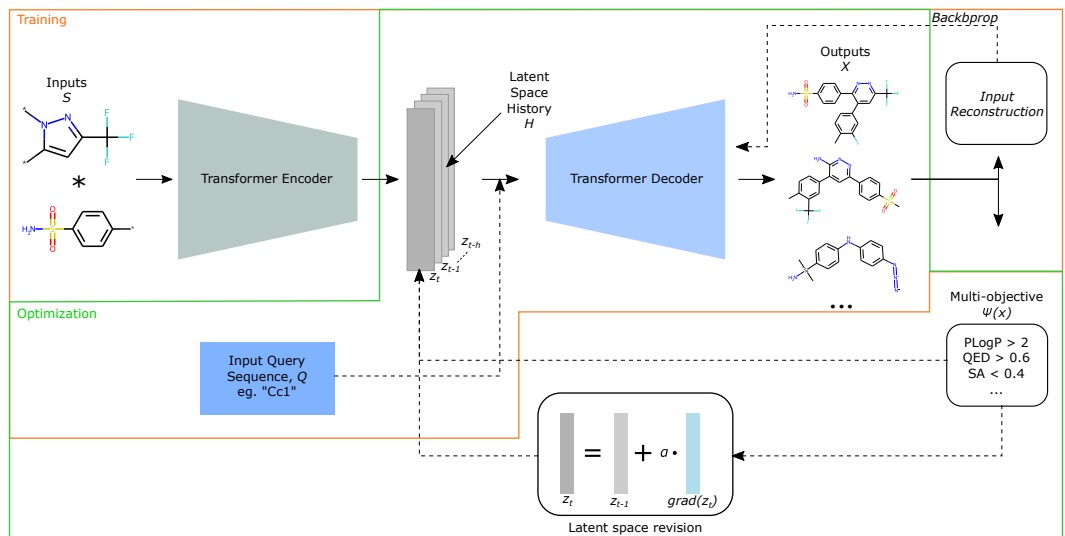

Figure 1: Overview of the SmilesFormer model and molecule optimization. The model is a VAE-based sequence-to-sequence Transformer. The inputs into the model can be full sequences, fragments, fragment compositions, or a dummy token. The input query sequence dictates the starting sequence for the output molecules.

learning rate constant. Our revised sequence $X^* = D(Q, z^*)$ is then obtained using the trained decoder. For scoring, we use a weighted sum of scores of individual properties $c_i$.

$$\Psi(x) = \frac{\sum w_i \times c_i(x)}{\sum_i w_i} \qquad (6)$$

Intuitively, the latent space manifold can be seen as a grid with nodes representing latent variables that satisfy certain multi-objective constraints. The optimization procedure, therefore, traverses this latent space towards a node that satisfies our chosen multi-objective properties. Each step of the latent space update produces variables that are judged whether to be better or worse than a previous step. Essentially, when a node has a better evaluation than a previous node, the latent variables are further updated in that direction. To halt the optimization, we introduce a maximum history constant. This is the number of previous nodes which can be revisited when a subsequent node gives a worse outcome. See algorithm 1. The process can also be halted by the maximum number of molecules expected to be generated.

### 3.3 MODEL INPUTS AND AUGMENTATION

Molecules are fed into the model by randomly choosing between two fragmentation strategies and a full sequence transformation. This helps to generate a wide range of input into the model and serves as input augmentation. In the case of a full sequence, the input canonical SMILES are transformed into non-canonical SMILES by changing the starting atom from which the SMILES string is formed. Note that while this is done, the canonical SMILES are expected as model output. This ensures that the model does not simply memorize the mapping between the input and output but is able to learn the intrinsic vocabulary of the SMILES strings. Examples of the SMILES transformation are shown in figure 2. In the case of fragmentation, we use two strategies as follows;

- Fragment-on-bonds: This strategy breaks down a molecule at the single bonds, resulting in fragments that indicate functional groups without the characteristic elements (Wüthrich et al., 2021). The intuition is that this should help the model to learn the general structure of a molecule without specific functional groups and atoms.

---

**Algorithm 1** Molecule Optimization Algorithm

---

Seed sequence $x_0 \in X$
Query sequence $q \in Q$
Learning rate $\alpha \in (0, 1)$
Latent vectors $z_0 \leftarrow f(x_0, p(Z))$
History $H \leftarrow [z_0]$
score $Y \leftarrow 0$
**while** $len(H) \neq 0$ **do**
    $x_i \leftarrow g(z_i, q)$
    $y_i \leftarrow \Psi(x_i)$
    **if** $y_i \geq Y$ **then**
        $Y \leftarrow y_i$
        $z_i \leftarrow z_i + \alpha \times grad(z_i)$
        $H \leftarrow [z_i, ... : H_{max}]$
    **else**
        $z_i \leftarrow H.pop()$
    **end if**
**end while**

---

- Recap decomposition: Based on a function supplied by RDKit[1](Lewell et al., 1998), this strategy uses a set of chemical transformations mimicking common reactions carried out in the lab in order to decompose a molecule into series of reasonable fragments. This kind of fragment is more complete as it includes characteristic elements and structures that form a functional group in the molecule. See fragmentation examples in figure 2.

Finally, the inputs are tokenized as in Schwaller et al. (2019) where each part of the SMILES string is represented by a single token allowing any arbitrary molecule to be transformed into the model input. To allow fragments composition in which multiple fragments can be fed into the model, we introduced a new token, *[COMP]*, as in *[\*]c1ccc(S(N)(=O)=O)cc1[COMP][\*]c1cc(C(F)(F)F)nn1[\*][COMP][\*]c1ccc(C)cc1*. This is the same as combining the fragments in figure 2(a) into a single model input.

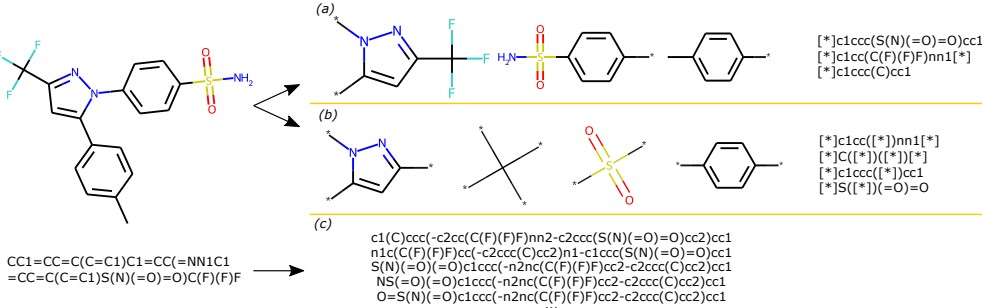

Figure 2: Model inputs and augmentation. The canonical SMILES are randomly fragmented by (a) Recap decomposition or (b) Fragment-on-bonds, or transformed into a (c) non-canonical full sequence. The *[\*]* in the fragments signify connection points.

## 3.4 MODEL STOCHASTICITY AND LATENT SPACE EXPLORATION

Dropout was introduced as a simple way to prevent neural networks from overfitting (Srivastava et al., 2014). However, it has recently been used as a Bayesian approximation to represent model uncertainty (Gal & Ghahramani, 2016), showing that using dropout is mathematically equivalent to an approximation to the probabilistic deep Gaussian process. In this work, we utilize dropout to introduce stochasticity in the generator model thereby allowing us to explore multiple decodings

---

[1]https://www.rdkit.org/

of a single latent space vector. This means that each vector can be sampled severally to produce different output sequences. To our knowledge, the only other work in which a similar idea has been explored is in the use of dropout as auxiliary exploration to reinforcement learning for continuous action problems (Sung et al., 2018).

### 3.5 TRANSFER LEARNING FOR SPECIFIC TASKS

To apply the trained model to specific tasks, the model is pre-trained with a generative capacity and the potential to sample compounds from a vast area of the available chemical space. This pre-trained model becomes a prior for training for specified tasks over a smaller set of compounds that are relevant for the given task. For example, if we aim to optimize generated molecules for a certain property, we would curate a dataset of molecules that were evaluated for this property. This will result in a model that produces molecules similar to the target dataset with a higher probability. Subsequently, generated molecules can be scored in order to sample the most promising molecules for the chosen target. We demonstrate this pipeline in section 4.2.

## 4 EXPERIMENTS

We carry out experiments on the proposed model with various *de novo* molecular design tasks. Evaluations are based on previously proposed tasks (Jin et al., 2020b; Nigam et al., 2021; Fu et al., 2022). We also demonstrate a drug rediscovery pipeline for donepezil, a known acetylcholinesterase inhibitor (Sugimoto et al., 2002). Our model is trained on the ChEMBL dataset (Mendez et al., 2018), containing 1.9 million molecules. Actual training molecules were limited to a molecular weight of 900 g/mol making the trained model more applicable to a small molecules setting. To encourage diversity in the generated molecules, query tokens were obtained from the training dataset by first tokenizing the molecules, then selecting a certain number of start tokens. The model generates molecules based on the query tokens by predicting the remaining sequence.

### 4.1 MULTI-OBJECTIVE *de novo* DESIGN

We consider the following target molecular properties: (1) Quantitative estimate of drug-likeness (QED) (Bickerton et al., 2012); (2) Octanol-water partition coefficient (LogP) (Kusner et al., 2017), an indicator for the molecule's solubility in water; (3) Synthetic Accessibility (SA), which is a measure of how hard or easy it is to synthesize a given molecule; (4) Inhibition against c-Jun N-terminal Kinases-3 (JNK3), belonging to the mitogen-activated protein kinase family, and are responsive to stress stimuli, such as cytokines, ultraviolet irradiation, heat shock, and osmotic shock; and (5) Inhibition against Glycogen synthase kinase 3 beta (GSK3$\beta$), an enzyme associated with an increased susceptibility towards bipolar disorder. (4) and (5) are evaluated using predictive models trained on positive and negative compounds according to Li et al. (2018). We conduct molecules generation that optimizes the mean value of "JNK3+GSK3$\beta$" and "QED+SA+JNK3+GSK3$\beta$" and compare our results against state-of-the-art generative methods; (1) Molecule Deep Q-Network (MolDQN) (Zhou et al., 2019), which combines state-of-the-art reinforcement learning techniques with chemistry knowledge to optimize molecules; (2) RationaleRL (Jin et al., 2020a), which builds a vocabulary of substructures to be expanded into full molecules; (3) Markov Molecular Sampling (MARS) (Xie et al., 2021), which employs Markov chain Monte Carlo sampling on molecules,; (4) Differentiable Scaffolding Tree (DST) (Fu et al., 2022), which utilizes a learned network to convert discrete chemical structures to locally differentiable one; and (5) Parallel Tempered Genetic Algorithm Guided by Deep Neural Networks (JANUS) (Nigam et al., 2021), which propagates two populations, one for exploration and another for exploitation, improving optimization by reducing expensive property evaluations.

The results show that SmilesFormer performs comparatively with the state-of-the-art approaches, outperforming on metrics such as success rate and novelty. Our model can also find the best scoring molecules in all the tasks except for QED+SA+JNK3+GSK3$\beta$. SmilesFormer, however, does not perform as well on the diversity metric. This could be primarily due to starting sequences, which may lead to many similar molecules being generated for the same starting sequence. Qualitatively, molecules from SmilesFormer tend to be much simpler compared with other methods. We believe this

Table 1: Results of Multi-objective *de novo* design

| Method | JNK3+GSK3$\beta$ | | | | QED+SA+JNK3+GSK3$\beta$ | | | |
|---|---|---|---|---|---|---|---|---|
| | Best | SR | Nov | Div | Best | SR | Nov | Div |
| MolDQN | 0.46 | 0.028 | **1.000** | 0.605 | 0.45 | 0.127 | **1.000** | 0.599 |
| RationaleRL | 0.81 | **1.000** | 0.973 | 0.824 | 0.75 | 0.748 | 0.568 | 0.701 |
| MARS | 0.78 | 0.802 | 0.979 | 0.699 | 0.72 | 0.734 | 0.984 | 0.714 |
| JANUS | - | **1.000** | 0.744 | 0.877 | - | **1.000** | 0.184 | 0.841 |
| DST | 0.89 | 0.920 | 0.975 | 0.745 | **0.83** | 0.798 | 0.988 | 0.774 |
| SmilesFormer | **0.90** | **1.000** | **1.000** | 0.715 | 0.805 | **1.000** | **1.000** | 0.726 |

Table 2: Results of single objective *de novo* design

| Method | JNK3 | | | | GSK3$\beta$ | | | |
|---|---|---|---|---|---|---|---|---|
| | Best | SR | Nov | Div | Best | SR | Nov | Div |
| MolDQN | 0.64 | 0.63 | 0.63 | 0.54 | 0.53 | 0.53 | 0.53 | 0.53 |
| RationalRL | 0.97 | 0.97 | 0.97 | 0.95 | 0.95 | 0.95 | 0.95 | 0.95 |
| MARS | 0.92 | 0.91 | 0.90 | 0.95 | 0.93 | 0.92 | 0.93 | 0.92 |
| JANUS | - | **1.000** | 0.744 | 0.877 | - | **1.000** | 0.184 | 0.841 |
| DST | 0.97 | 0.97 | 0.97 | 0.95 | 0.95 | 0.95 | 0.95 | 0.95 |
| SmilesFormer | **0.99** | **1.000** | **1.000** | 0.737 | **0.99** | **1.000** | **1.000** | 0.725 |

is because the model outputs are constrained due to the learned association with the input fragments. Hence, generated molecules are easily decomposable into individual fragments. See figures 6 and 7.

### 4.1.1 ORACLE EFFICIENCY

Based on previous works, realistic molecule optimization tasks can be time-consuming and expensive, therefore it is necessary to evaluate the efficiency of oracle calls needed for a generative model to reach the multi-objective goal. We follow the example in Fu et al. (2022) to compare the oracle efficiency of our approach to other state-of-the-art methods, shown in Figure 3. Compared to other methods, our method showed a significant advantage by reaching optimum performance with much fewer oracle calls. By leveraging model parallelization, our model can suggest multiple candidates per iteration thereby speeding up the optimization process.

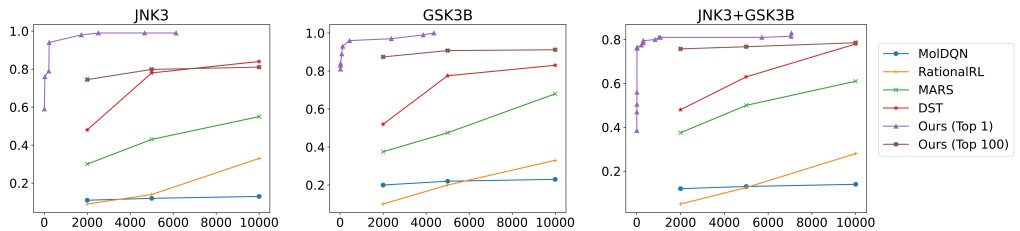

Figure 3: Oracle efficiency test. Top-100 average score v.s. the number of oracle calls. Top 1 results for Smilesformer are also shown. The results have been limited to 10000 molecules since our model reaches optimal molecules much faster than other methods.

### 4.2 DONEPEZIL REDISCOVERY

In this section, a drug discovery pipeline is demonstrated using our model and optimization strategy. Our goal was to evaluate the capability of the presented approach in generating novel scaffolds with complex molecular structures not represented in the training data. As a protein target, we selected

acetylcholinesterase (AChE), which has been successfully targeted to improve cognitive function in Alzheimer's disease with molecules such as tacrine, physostigmine, rivastigmine, galantamine, and donepezil that exhibit diversity in molecular scaffolds (Marucci et al., 2021). This allowed us to remove data associated with one of the chemotypes (i.e donepezil) from the training data to test our approach while having enough data for training. The removed data was used for evaluating if SmilesFormer can generate and rank as top suggestions any of the hidden known potent AChE inhibitors. Successful generation of such molecules would indicate the ability of the model to reach the potent novel scaffolds.

Due to the retrospective nature of the problem setting, one critical aspect of data preparation was to accurately remove molecules associated with the donepezil scaffold from both pretraining and training datasets. First, the pretraining dataset (ChEMBL database Gaulton et al. (2011)) was filtered to exclude all molecules containing donepezil scaffold (Figure 4a). This ensures that our model does not simply memorize the target molecule and output it in the generation step. We further trained the model with AChE inhibitors curated from ChEMBL database. This dataset consists of 3950 molecules with IC50 activity measurements and was also filtered to exclude the donepezil scaffold and other relevant molecules containing major donepezil pharmacophores such as 1-indanone, the simultaneous presence of indan with either benzylpiperazine or benzylpiperidine (Figure 4b) since such a combination will have a high similarity to the pursued donepezil scaffold and make the problem setting trivial.

During generation step, a multi-objective function was used to optimize the generated molecules with the following properties; (1) SA $C_i < 1, Target = 0$; (2) QED $C_i > 0, Target = 1$; (3) PLogP $C_i > 0, Target = 40$; (4) Molecular Weight $250 < C_i > 750$; (5) Number of Hydrogen Bonds $5 < C_i > 10$; (6) Topological Polar Surface Area (TPSA) $75 < C_i > 150$; (7) Number of Rotatable Bonds $5 < C_i > 10$; (8) Similarity to tacrine $C_i < 1, Target = 0$; (9) Similarity to physostigmine $C_i < 1, Target = 0$ (10) Similarity to rivastigmine $C_i < 1, Target = 0$; and (11) AChE inhibitor activity pIC50 (negative log IC50), determined by a trained predictive model on AChE Inhibitor active compounds $C_i > 1, Target = 1$. Similarity assessment to tacrine, physostigmine, and rivastigmine molecules was used to direct the model towards exploring novel scaffolds not represented in the training data, since discovering novel molecules not documented before is of a high priority in drug discovery due to patentability concerns. Up to 30,000 molecules were generated and ranked according to the aggregate generative score.

In a post-processing step, the generated molecules were filtered using a set of Medicinal Chemistry Filters (MCF) to remove molecules with undesirable toxic and unstable structures. Molecules were also assessed for novelty by filtering out molecules with (a) substructure match to tacrine, physostigmine, and rivastigmine scaffolds in order to avoid building on these scaffolds (b) scaffold match to the molecules in the training set, and (c) Tanimoto similarity score > 0.5 to the molecules in the training set. Moreover, AChE protein is found in the brain, thus successful molecules should be able to pass the blood brain barrier (BBB) and reach the target tissue to exert activity. Therefore, generated molecules were passed through a BBB permeability threshold of 0.99 based on a trained predictive model on BBB permeability. The final list of molecules is then clustered and ranked with a desirability score of normalized $(QED + pIC50)/2$. The generation and post-processing steps were run 10 times and the top 20 scaffolds from the combined runs were considered as the final result. The top 10 scaffolds are shown in figure 9. In the final set of molecules, we found the molecule shown in figure 4c, which was a late-stage intermediate molecule leading to the discovery of Donepezil (Sugimoto et al., 2002). The results show that our model is able to generate a molecule that contains the scaffold and important pharmacophores of a known potent drug, and ranks such a molecule high after an extensive post-processing pipeline. This result provokes a thought of a potential presence of other potent compounds among generated molecules that are yet to be tested experimentally.

### 4.3 EXPERIMENTS WITH OPTIMIZATION PARAMETERS

#### 4.3.1 DROPOUT AS EXPLORATION

In this experiment, the effect of dropout was examined on the performance of the exploratory capability of our model. The final model was trained with a dropout value of 0.15 and tested in the generation and optimization step with varying dropout values. We used the same objective as in section 4.2. It can be observed that increasing the dropout value results in the model being able to

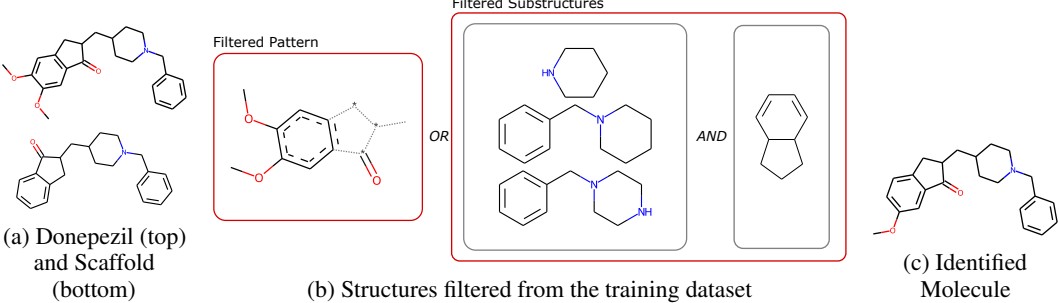

(a) Donepezil (top) and Scaffold (bottom)

(b) Structures filtered from the training dataset

(c) Identified Molecule

Figure 4: Donepezil rediscovery experiment pipeline.

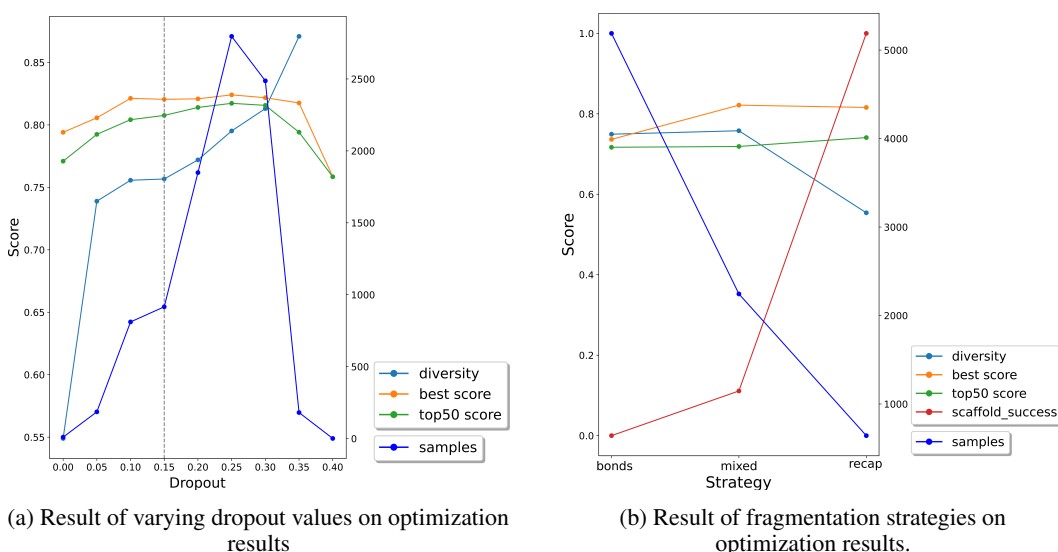

(a) Result of varying dropout values on optimization results

(b) Result of fragmentation strategies on optimization results.

Figure 5: Optimization parameters experiment result.

explore more latent space representations evidenced by the number of valid samples generated and the diversity of the molecules generated (see figure 5a). However, the model is seen to perform worse when the dropout value is increased above 0.25.

### 4.3.2 FRAGMENTATION STRATEGIES

In this experiment, the effect of the fragmentation strategies was examined. The model was tasked with optimizing a molecule from the donepezil rediscovery experiment. This was done by setting the molecule as input and applying combinations of the fragmentation strategies introduced in section 3.3. The mixed strategy (see figure 5b) randomly switches between the bonds fragmentation and recap fragmentation. The recap strategy achieves higher success in finding molecules with donepezil scaffold, however, the mixed strategy achieved a marginally higher best score and molecule diversity.

## 5 CONCLUSIONS

In this work, we introduced SmilesFormer, a language model for *de novo* molecule generation and optimization. Our model learns from a vast amount of training data to represent and explore chemical spaces in order to generate new compounds. We also introduced the optimization strategy, which systematically and more efficiently traverses the space of latent representation of the model to generate compounds that satisfy a multi-objective function. We demonstrated promising results for a variety of tasks with relevance to drug discovery and molecular optimization.

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

## A  APPENDIX

### A.1  TRANSFORMER MODEL

The transformer model is a stepwise autoregressive encoder-decoder model composed of a combination of multi-head attention layer and positional feed-forward layers (Schwaller et al., 2019). A multi-head attention layer computes the attention as follows:

$$attention(Q, K, V) = softmax\left(\frac{QK^T}{\sqrt{d_k}}\right) V \tag{7}$$

where $K$, $V$, and $Q$ are the key, value, and query of the layer. The sequential nature of the data is encoded with positional encoding to allow the network to know the relative position of tokens in a sequence:

$$PE_{pos,2i} = sin\left(\frac{pos}{10000^{\frac{2i}{d_{emb}}}}\right), PE_{pos,2i+1} = cos\left(\frac{pos}{10000^{\frac{2i}{d_{emb}}}}\right) \tag{8}$$

Our approach treats the transformer model as a variational autoencoder with the aim of learning smooth latent space representation of the input data.

### A.2  MODEL AND TRAINING CONFIGURATION

Dataset preparation: All preprocessing and preparation were performed online during training. In all our experiments the following rules were followed; (1) Maximum molecular weight of 900, to limit molecules to small molecules; (2) minimum token length of 8; (3) maximum token length of 104;

Hardware: Model training and molecular design inference were carried out on the Intel(R) Xeon(R) CPU e5-2698 v4 @ 2.20GHz processors, 256GB RAM, and Tesla V100-DGXS graphics cards with 32GB VRAM.

The model's hyperparameters are as follows: 12 transformer layers, 12 transformer heads, an embedding size of 288, and 1024 hidden units. Training was done with a batch size of 64 per GPU and optimized with the ADAM optimizer (Kingma & Ba, 2015) with an initial learning rate of 0.0001. To construct the input sequence during training, the two fragmentation strategies are randomly mixed with the fragment composition probability (the probability of composing 2 or more fragments as input) of 0.75, full sequence probability of 0.5, and dummy input probability of 0.05.

### A.3  ADDITIONAL EXPERIMENTS

#### A.3.1  SINGLE OBJECTIVE MOLECULAR DESIGN

We compare the performance of various methods on single-objective *de novo* molecular generation for optimizing QED and LogP scores respectively. The results are shown in Table 3. We have added results from EGEGL (Wüthrich et al., 2021), An Interpretability-augmented Genetic Expert for Deep Molecular Optimization. We show the top molecules in single, as well as multi-property objectives optimization in figure 6.

#### A.3.2  CORRELATION BETWEEN INPUT AND OUTPUT VALUES ON CHEMBL MOLECULES

Molecules in the ChEMBL dataset were used as input into the model to generate a new molecule. The relationship between the input and output molecule properties is shown in figure 8. We observe a somewhat weak correlation between input and output properties, and a visible group of outliers in properties like LogP and Molecular Weight.

### A.4  EXPERIMENT WITH STARTING SEQUENCE LENGTH

We demonstrate the ability of the model to suggest complete valid sequences from partial sequences. The model is supplied with partial sequences of molecules sampled from the AChE

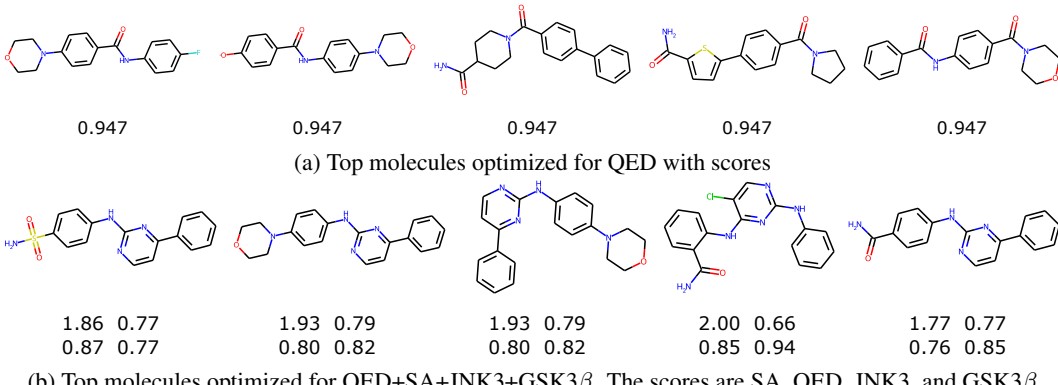

(a) Top molecules optimized for QED with scores

(b) Top molecules optimized for QED+SA+JNK3+GSK3$\beta$. The scores are SA, QED, JNK3, and GSK3$\beta$ respectively

Figure 6: Top molecules in single and multi-objective optimization tasks.

Figure 7: Fragment analysis for input and generated molecules. We observe the fragments in the generated molecules are also similar to the fragments recognized in the input molecules. The model utilizes the recognized fragments in the input molecules to generate new molecules optimized toward the objective function.

Table 3: Highest scores of generated molecules on single-objective *de novo* molecular generation

| Method | LogP | | | QED | | |
|---|---|---|---|---|---|---|
| | 1st | 2nd | 3rd | 1st | 2nd | 3rd |
| MolDQN | 11.8 | 11.8 | 11.8 | 0.948 | 0.948 | 0.948 |
| MARS | 45.0 | 44.3 | 43.8 | 0.948 | 0.948 | 0.948 |
| EGEGL | 34.6 | 34.4 | 34.4 | 0.948 | 0.948 | 0.948 |
| DST | 49.1 | 49.1 | 49.1 | 0.947 | 0.946 | 0.946 |
| SmilesFormer | 40.8 | 40.8 | 40.4 | 0.947 | 0.947 | 0.947 |

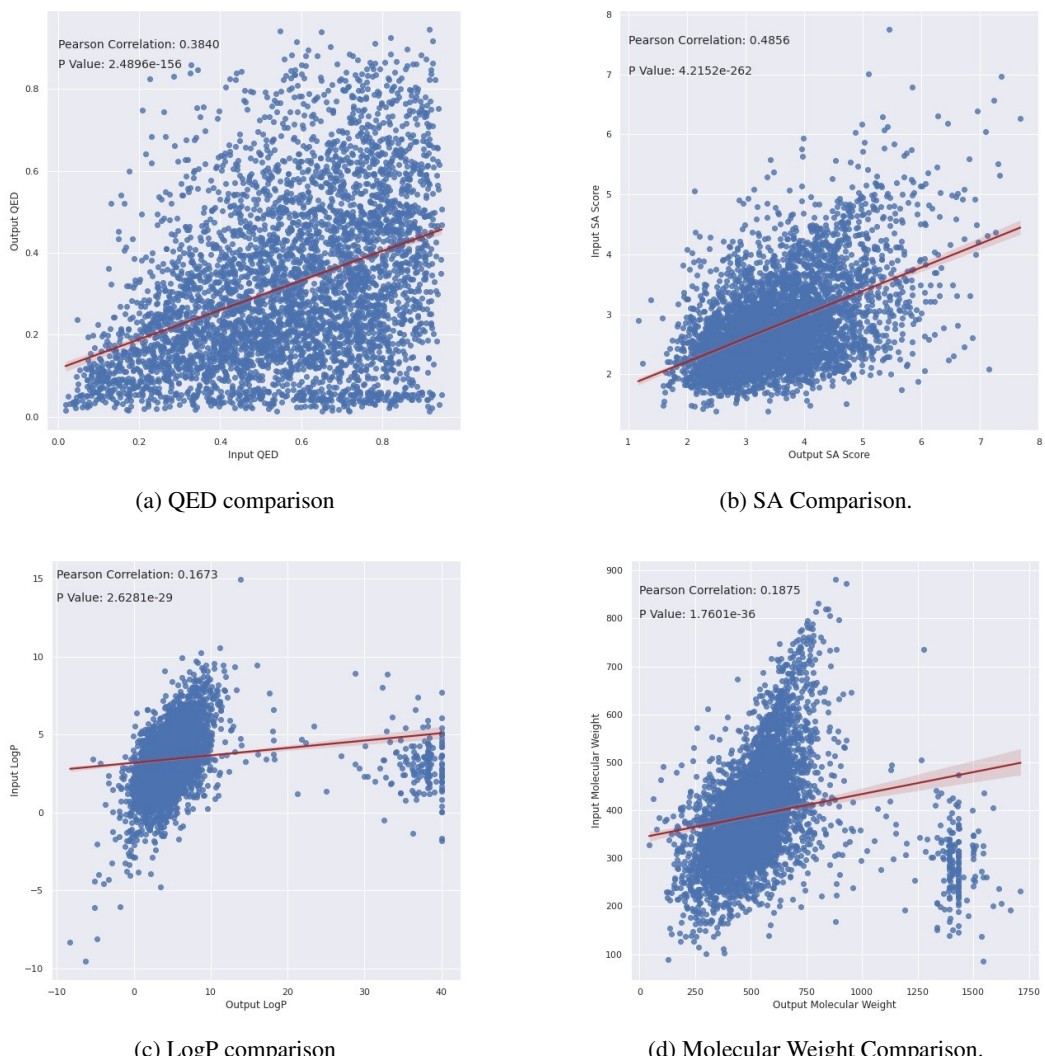

(a) QED comparison

(b) SA Comparison.

(c) LogP comparison

(d) Molecular Weight Comparison.

Figure 8: Correlation between input and output values on ChEMBL molecules.

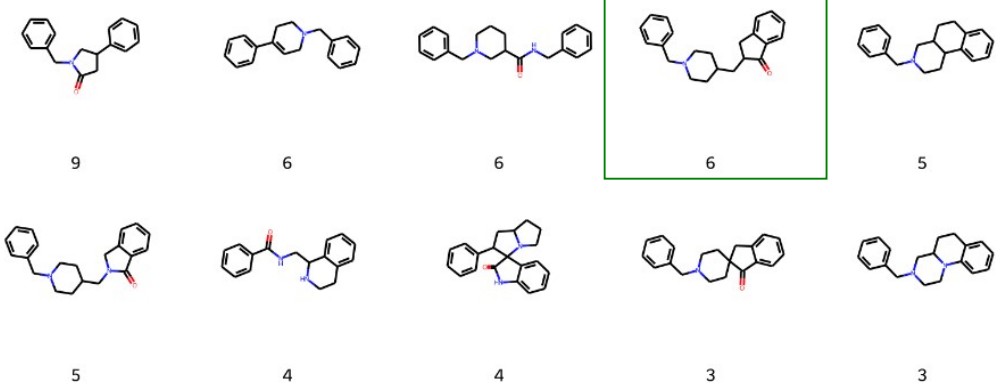

Figure 9: Top 10 scaffolds from the generated molecules. The numbers signify how many times the scaffold appears out of 10 experiment runs.

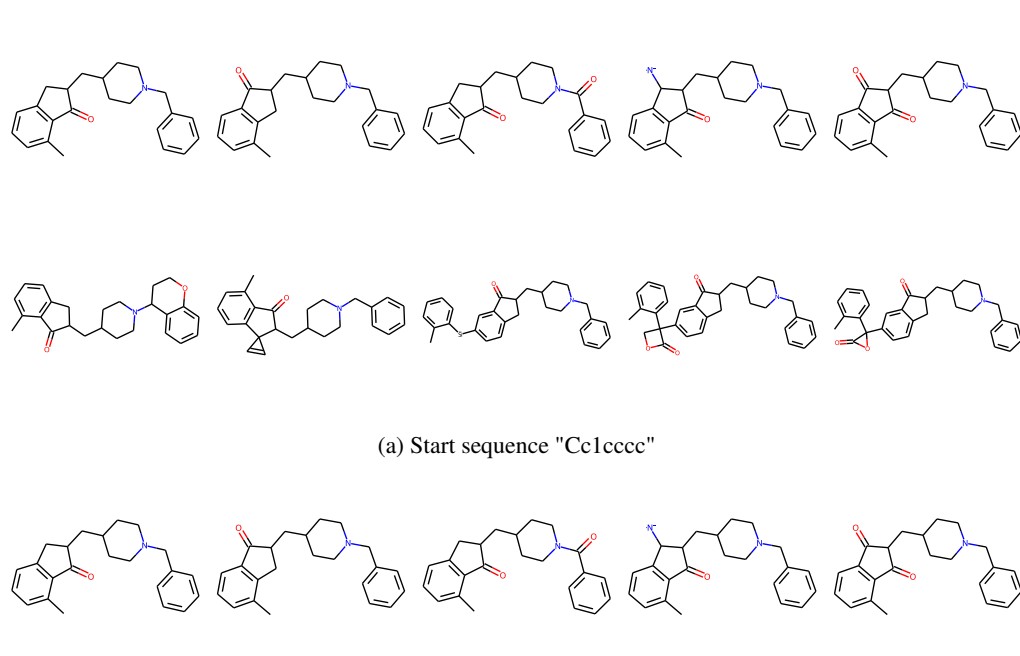

(a) Start sequence "Cc1cccc"

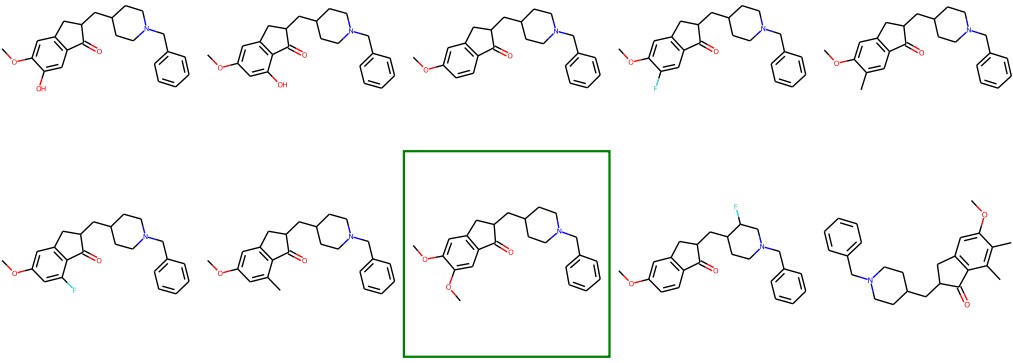

(b) Start sequence "CN(CCCN".

(c) Start sequence "COc1cc2". Donepezil is shown with green borders.

Figure 10: Molecules generated from input start sequences in a lead optimization task. The top 10 molecules from each experiment are shown. They all have the donepezil scaffold.

inhibitors as query sequences in a lead optimization task. The generated molecules are also optimized to be similar to a given scaffold. The multi-objective function is set up such that (1) Scaffold similarity $c_i > 0, Target = 1$; (2) Scaffold presence $c_i = 1$; (3) SA $c_i < 1, Target = 0$; (2) QED $C_i > 0, Target = 1$; (3) logP $-0.4 < C_i < 5.6$; The lead molecule is $COc1ccc2c(c1)C(= O)C(CC1CCN(Cc3ccccc3)CC1)C2$, an intermediate molecule in the donepezil discovery pipeline. Results from the top 3 partial sequences are shown in figure 10.

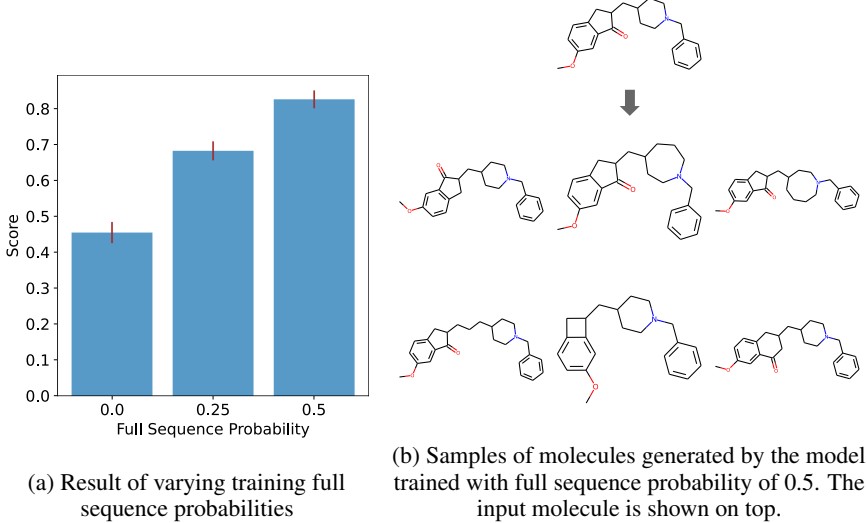

(a) Result of varying training full sequence probabilities

(b) Samples of molecules generated by the model trained with full sequence probability of 0.5. The input molecule is shown on top.

Figure 11: Experiment results for training full sequence probability.

## A.5 EXPERIMENT WITH FULL SEQUENCE PROBABILITY

We investigate the effect of the full sequence probability parameter on the model's performance in generating similar molecules as the input. In a lead optimization task, the model is expected to explore the chemical space of a lead molecule to generate similar molecules which could possess better properties than the lead molecule. In this experiment, the model is tasked with generating similar molecules to *COc1ccc2c(c1)C(=O)C(CC1CCN(Cc3ccccc3)CC1)C2* after being trained with varying full sequence probabilities. Results are presented in figure 11.

