# OpenReview forum: "SmilesFormer: Language Model for Molecular Design"
_ICLR.cc/2023/Conference — Submitted to ICLR 2023_

### Official Review · Reviewer_zCNZ · 2022-10-23

**Confidence:** 4
**Correctness:** 2
**Technical Novelty And Significance:** 2
**Empirical Novelty And Significance:** 2
**Recommendation:** 1

**Clarity, Quality, Novelty And Reproducibility:**


Clarity: the introduction and related works sections describe well the paper as a whole and the difficulties of the problem that the authors try to tackle. The description of the specifics of the model should be improved as explained above. There is not clarity about the evaluation, except for the donepazil rediscovery process.

Quality: SmilesFormer is not properly evaluated against other methods in the state-of-the-art. The current evaluation shows that the representation learning has some promise, but it is not enough as substance for all claims in this paper. This is aggravated by the notation and overall lack of clarity of the paper, which render the reproducibility of this work almost nonexistent.

Originality: expanding on what was explained above on weaknesses, the parts that works as functional parts, such like the use of VAE for reconstruction of smiles and the use of gradient descent for molecule optimization of the method are not novel.


**Strength And Weaknesses:**

Strengths
- the process of data-augmentation by using different strategies of smile sequence fragmentation or modification but still aiming at reconstructing valid canonical smile sequences is very interesting
- the introduction and related works sections are well written, it explained well the overall functional parts of the model and also have interesting points about the challenges for molecule design using generative models.

Weaknesses
- the description of the method should be improved, I had to re-read it multiples times before I was able to understand it. I believe that with a clear division of the description about how the diverse parts of the model are trained and another section with description about how the molecule generation process works could improve the readability of this paper.
- the paper lacks proper descriptions of the datasets used for the experiments
- This work cannot be reproduced. It lacks descriptions about how the experiments were run. The "model and training" section from the appendix should be expanded and moved to the main text.
- there are references for the evaluation, but the main text should at least briefly explain the evaluation measures. Now, it’s not easy to see why these measures show that the method performs well.
- although the method architecture is interesting, none of its parts are novel, authors should highlight how their modifications to existing parts of the model compare to the original papers. In particular, the multi-objective traversal of the latent space is not properly cited, but it is not novel for protein and drug engineering, this is not clear in the current version of this paper.
- the donepezil rediscovery is interesting, but the effectiveness of the model can’t be derived from a single case study. I think that process could have been replicated with other drugs in order to give more confidence to the rediscovery evaluation.


**Summary Of The Paper:**

The paper presents SmilesFormer, a transformer based method that aims at generating novel molecules with a multiple properties objective. This is achieved by training a variational auto-encoder fed with fragments of molecules that are then encoded into an embedding space. Afterwards, they are decoded to reconstruct the molecule by generating a valid canonical smile sequence. In order to obtain more desirable properties for the molecule, its representation is optimized in the embedding space by maximizing a score that represents the required properties. The authors claim that the model performs well against state-of-the-art methods.


**Summary Of The Review:**

It is clear the authors understand the problem at hand, but the paper lacks a clear description about the experiments that were performed to validate the claims. The explanation of the proposed model still needs work to be understandable and not let the readers make assumptions about how the molecule generation process works. The evaluation of the model is not clear, and it is not enough to sustain all the claims made by the author.

---

### Official Review · Reviewer_US7Z · 2022-10-24

**Confidence:** 4
**Correctness:** 2
**Technical Novelty And Significance:** 2
**Empirical Novelty And Significance:** Not applicable
**Recommendation:** 3

**Clarity, Quality, Novelty And Reproducibility:**

**Clarity**

The paper is well written and easy to read.

**Quality**

Overall, the paper is of good quality. However, I would have expected that the proposed model was tested in a standard benchmark such as GuacaMol or MOSES. The explanation provided by the authors for not doing so (quoting: "the task of molecular design is usually related to specific targets, earlier proposed benchmarks do not fully represent how a generative model is utilized in the drug discovery pipeline") is not convincing since most of the works cited in this paper (all generative models that do molecular optimization) do provide a benchmark evaluation.

**Novelty**

Extremely limited. The idea of using Transformer-based models to learn a language model of SMILES is not new [1]; the use of a fragment-based approach for SMILES strings is itself not new ([2] uses a very similar strategy to decompose a molecule SMILES into fragments using the BRICS algorithm); the idea of using gradient ascent to optimize the molecular property in latent space is strikingly similar to [3]. All of these works are not listed in the background part, and the differences with the proposed approach are not discussed.

**Reproducibility**
Not reproducible (no code is provided).

[1] Irwin et al., _Chemformer: a pre-trained transformer for computational chemistry_. Mach. Learn.: Sci. Technol. 3 (2022)

[2] Podda et al., _A Deep Generative Model for Fragment-Based Molecule Generation_. AISTATS (2020)

[3] Jain et al., _Multi-target optimization for drug discovery using generative models_. ICML workshop on Computational Biology (2021)

**Strength And Weaknesses:**

**Strengths**
- Works well in multi-property optimization scenarios
- Very efficient in terms of oracle calls
- Re-discovering of Donepezil looks impressive.

**Weaknesses**
- not evaluated on standard benchmarks such as GuacaMol, MOSES
- limited novelty
- literature not fully discussed

**Summary Of The Paper:**

The paper describes a transformer-based language model for SMILES strings. The model is a VAE trained on entire molecules, molecular fragments, or their composition. It can be used for molecular (multi-)property optimization with a stochastic latent space traversal technique. The experiments show that it can be applied with success on multiple property optimization, as well as in real-world drug design scenarios (in particular, the authors apply the method to rediscover Donepezil).

**Summary Of The Review:**

In summary, this seems to be a borderline paper. However, at the moment the weaknesses greatly overcome the strengths, so I'm forced to recommend rejection.

Besides addressing the points raised above (see Quality/Novelty), I'd like the authors to answer/comment on the following questions/comments in order to let me be able to eventually raise my score:

- What are the valid/unique fractions of the molecules generated in tables 1 and 2?
- Can you show the distribution of molecular features (such as the number of rings, bonds, atoms, etc.) of the generated molecules in comparison to the molecules in the test set?
- Can you provide an explanation as to why the model seems to perform not so well in the logP optimization task? (see Table 3)

---

### Official Review · Reviewer_TxfB · 2022-10-25

**Confidence:** 3
**Correctness:** 3
**Technical Novelty And Significance:** 3
**Empirical Novelty And Significance:** 3
**Recommendation:** 5

**Clarity, Quality, Novelty And Reproducibility:**

Figure 1: the dotted lines cross one another, making it hard for the reader to easily discern the flow. Features such as words and asterisks on molecules are too small to see.
Table 2: it would be clearer if the authors include the meaning of SR, Nov and Div, which I assume mean ‘success rate’, ‘novelty’ and ‘diversity’ respectively.


**Strength And Weaknesses:**

Strengths:
Proposed methods showed competitive performances compared to existing baselines
The fragment-based input augmentation is a novel approach to enhancing molecular representations in molecular deep learning.
Weaknesses:
The experimental section on “Donepezil rediscovery” relies on many post-processing steps on top of the generation with SmilesFormer and generates as many as 30k candidates for the Donepezil scaffold to be discovered. Without comparison with a suitable baseline in this section, it is hard for the reader to appreciate the impact of this section.


**Summary Of The Paper:**

The paper proposes SmilesFormer, a transformer-based model that learns a latent space between its encoder and decoder module. The model is trained with 2 main objectives: a) s molecule reconstruction objectives where the model takes in fragments of a molecule and learn to reconstruct its original structure and b) a KL divergence loss between its learned latent space and a normal distribution prior. Input augmentations, such as non-canonical SMILES and fragments made through the Recap decomposition or fragment-on-bonds approaches, were used to better learn molecular representations through reconstructions. During inference, the model can take in an input query and generate a modified variant through gradient step backpropagated from oracle models that predict the target properties of the generated molecules, with stochasticity introduced through dropout. The authors conducted experiments to show that the SmilesFormer can outperform baseline methods in several instances in the applications of multi-objective de novo design with better oracle efficiency.

**Summary Of The Review:**

The paper studies an important application of de novo molecular design and proposes novel contributions such as the fragmentation-based optimization. The other parts of the SmilesFormer mostly relies on existing approaches. My main concern about this manuscript is the “Donepezil rediscovery” section, as mentioned in the weakness, where much post-processing is involved and many candidates are generated for the compound of interest to be discovered. A comparison with baselines in this section would help the reader to better appreciate the model’s impact.

---

### Decision · Program_Chairs · 2023-01-20

**Decision:**

Reject

**Justification For Why Not Higher Score:**

All reviewers vote for rejection.

**Justification For Why Not Lower Score:**

N/A

**Metareview: Summary, Strengths And Weaknesses:**

Summary:

The paper describes a transformer-based language model for SMILES strings. The model is a VAE trained on entire molecules, molecular fragments, or their composition. It can be used for molecular (multi-)property optimization with a stochastic latent space traversal technique. The experiments show that it can be applied with success on multiple property optimization, as well as in real-world drug design scenarios (in particular, the authors apply the method to rediscover Donepezil).

Strengths:

- Proposed methods shows competitive performances compared to existing baselines.
- The fragment-based input augmentation is a novel approach to enhancing molecular representations in molecular deep learning.
- Works well in multi-property optimization scenarios
- Very efficient in terms of oracle calls
- Re-discovering of Donepezil looks impressive.
- data-augmentation by using different strategies of smile sequence fragmentation or modification but still aiming at reconstructing valid canonical smile sequences is very interesting

Weaknesses:

- The experimental section on âDonepezil rediscoveryâ relies on many post-processing steps on top of the generation with SmilesFormer and generates as many as 30k candidates for the Donepezil scaffold to be discovered. Without comparison with a suitable baseline in this section, it is hard for the reader to appreciate the impact of this section.
- not evaluated on standard benchmarks such as GuacaMol, MOSES
- literature not fully discussed

- The description of the method should be improved.
- the paper lacks proper descriptions of the datasets used for the experiments
- This work cannot be reproduced. It lacks descriptions about how the experiments were run.
- there are references for the evaluation, but the main text should at least briefly explain the evaluation measures.
- the donepezil rediscovery is interesting, but the effectiveness of the model canât be derived from a single case study.

Recommendation:

All reviewers agree on rejection. I, therefore, recommend to reject the paper. I encourage the authors to use the feedback provided
by the reviewers to improve the paper and then resubmit to another venue.